# Retrieving Intangibility, Stemming Biodiversity Loss: The Case of Sacred Places in Venda, Northern South Africa

**Innocent Pikirayi *** and **Munyadziwa Magoma**

Department of Anthropology and Archaeology, Faculty of Humanities, University of Pretoria, Pretoria 0028, South Africa; u11311828@tuks.co.za
* Correspondence: innocent.pikirayi@up.ac.za

**Abstract:** Sacred sites and landscapes mirror indigenous peoples' identity, well-being and sense of place. In Venda, northern South Africa, such places are preserved through myths and legends. Following a scoping study, which also involved engagement with indigenous communities, we reveal how human-driven destruction of biodiversity contributes towards significant losses of such heritage. Large-scale agriculture, mining and commercial plantations around Thathe forest, Lake Fundudzi and Phiphidi waterfalls are not only destroying these places, but also impoverishing indigenous peoples. This is not sustainable from the perspective of heritage conservation, survival and well-being of indigenous communities. Recognising intangible values expressed through myths and legends is necessary in heritage conservation and in addressing some of the Sustainable Development Goals (SDGs).

**Keywords:** biodiversity loss; Venda; cultural landscape; myths; legends; Thathe forest; Lake Fundudzi; Phiphidi waterfalls





## 1. Introduction and Research Context

Sustainable Development Goals (SDGs) are about the future, and how any generation perceives such [1–3]. A French Jesuit priest, scientist, paleontologist, theologian, philosopher and teacher, Pierre Teilhard de Chardin (1881–1955) once said; "The future belongs to those who give the next generation reason for hope". Pierre Chardin was Darwinian, and the writer of several influential theological and philosophical books [4,5]. Though considered by his peers as controversial and in some cases his works condemned by the Roman Catholic Church's Congregation for the Doctrine of the Faith, we however appraise his approach to palaeontology, which, in our view, were far ahead of his time. Pierre Chardin was involved in the discovery of Peking Man (*Homo erectus pekinensis*)—a subspecies of *H. erectus*—at Zhoukoudian, in northern China during the mid-1920s [6].

As a Darwinian scholar, Teilhard de Chardin perceived the universe in evolutionary terms, seeing it constantly shifting towards a state of greater complexity and higher levels of consciousness. Within this evolutionary process, he discerned several changes, or transitions. According to Teilhard, the beginnings of life on earth and the emergence of human consciousness are two critical thresholds in this process. These changes or transformations trigger new stages in a continuous process of development. He saw the world as a single continuous process of development—what he referred to as "universal interweaving"—with diverse levels of organization and each entrenched in earlier levels and its emergence seen as the actualization of what was potentially present in earlier levels. According to Teilhard de Chardin there was no sharp divide in the word between humanity and other animals. Instead, he saw a single evolving entity with mutually interconnected events, naturally progressing from matter to life to human existence to human society. In this paper, we adopt this perspective in our perception of biodiversity—the biological variety and variability of life on earth—as a measure of variation at the genetic, species, and ecosystem level that is now primarily human driven. As some scholars have demonstrated,

population size and growth, along with overconsumption, are significant factors in biodiversity loss and soil degradation [7]. They warn of a "ghastly future of mass extinction, declining health and climate-disruption upheavals" that threaten human survival because of ignorance and inaction [7]. One cannot not detach humans from biodiversity as humans have biological origins and are species in themselves. Even the World Heritage Convention *Operational Guidelines* [8] acknowledge this in the identification of sites and landscapes for world heritage listing.

This paper is based on research among the Venda people of northern South Africa (Figure 1). The area in which they live is also referred to by the same name. Venda people reside on the north-eastern foothills of the Soutpansberg mountains [9–11]. The research shows that there are consequences when intangible values ascribed to sacred landscapes are ignored. There is value in preserving and conserving intangible heritage [12,13]. Sacred sites and landscapes in Venda are slowly being eroded and decimated by modern development and intensifying human settlement.

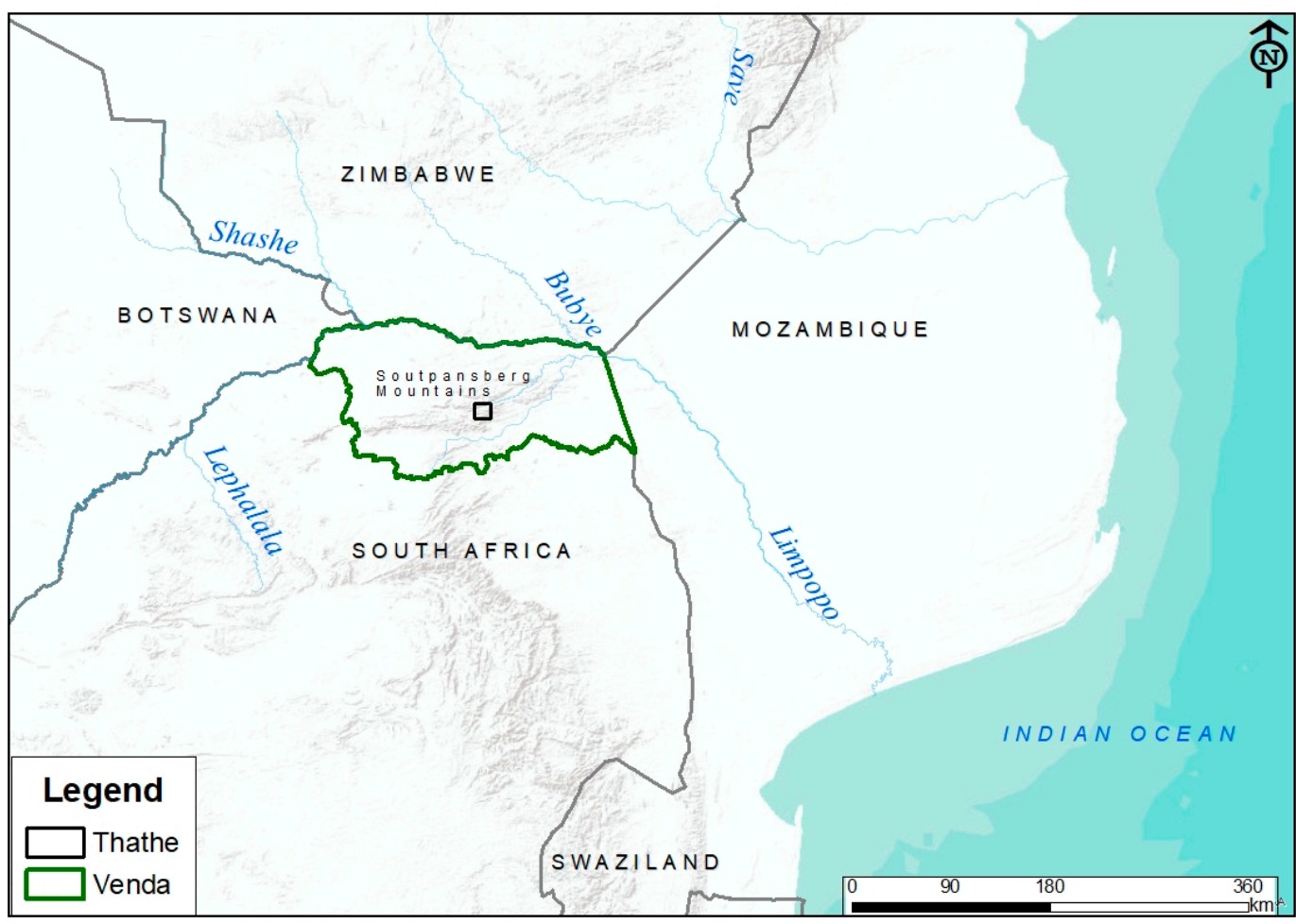

**Figure 1.** The location of Venda, northern South Africa. The research focuses on Thathe and surrounding areas on the foot of the Soutpansberg mountains (Map, prepared by Brenda Makanza).

The Venda are historically connected to some royal palaces, sacred sites and cultural landscapes found in the area [14]. The area is well-watered, moist and fertile, but may also be extremely hot and dry. The Venda are not a monolithic ethnic group, but rather a "composite" entity, comprising culturally different, but related groups or clans. Their origins are discussed in oral traditions [15–19], history and archaeology [15,20–24], all these sources identifying them as descendants of many heterogeneous groups and clans, including the Singo, Ramabulana, Tshivhase, Mphephu, Mphaphuli, Ngona, Nelwamondo and Nethengwe. The Ngona and Mbedzi are considered original inhabitants of Venda, while the Twanamba, Nyai, Tavhatsindi, Lembethu, Singo, Vhalaudzi (Vhasenzi) and Lemba are regarded as 'outsiders'. However [24], these groups may have been formed by some elites, to take advantage of feuds between royal houses. Some Venda claim origins from the northern part of the southern African sub-continent and seem to have interacted with Sotho-speakers from south-eastern Africa [16]. Although oral traditions acknowledge smaller infiltrations to the Soutpansberg region coming from the north of Limpopo River, two migrations stand out. The Ngona and Tavhatsindi clans arrived first, followed by the Singo, who became rulers of the Venda people [25,26]. The land of Ngona was later settled by Karanga-Rozvi clans from Zimbabwe, post 1700 AD [27]. They briefly conquered the Venda and set up a state ruled by the Singo dynasty. These migrations are attested in Venda language, which has elements of both Sotho and Karanga [10,11].

## 2. Research Questions, Aims and Objectives, and Methodology

Our key research question is the degree to which cultural heritage losses triggered by the current global environmental crisis have negatively impacted on indigenous communities [28]. Beyond heritage and development actors, we sought to understand the extent to which such conversations may be held with communities whose heritage they seek to conserve.

The aim of our research project is to assemble data that relates to cultural heritage and biodiversity conversation in Venda. Of particular focus is the value of intangible heritage in helping conserve tangible heritage—in our case, sacred forests, lakes and other landscapes. The destruction of tangible cultural heritage in Venda, just like other places in the world, is also due to lack of or poor appreciation of the intangible values associated with the former. The objective of the project is to understand the connections between cultural heritage and biodiversity conservation in this part of Africa and how these may contribute to the global discussion on sustainable development goals. One of us is (IP) is involved in a project on how universities are contributing towards addressing sustainable development goals. For the Venda, we aim to share the data with traditional leadership and elders, to highlight the value of sacred sites and landscapes and the necessity of conserving them.

Our data was initially collected as a desktop study on indigenous Venda histories and how various clans in northern South Africa construct and shape their worldview. It became evident during the desktop exercise that the Venda worldview was increasingly lost to modern development, due to the destruction of sacred forests, sacred waterscapes and shrines. Modern development in this area goes back to pre-1994, when South Africa was under apartheid rule, characterized by serious racial and other forms of colonial oppression and segregation. Apartheid rule has ushered a legacy of gross social and economic inequalities in contemporary South Africa, where development projects are imposed on local and indigenous communities, with promises to eradicate poverty and improve their well-being. Concerned with the pace of such developments, we consulted with some traditional leaders, site custodians and community elders to assess the impact and sustainability of commercial forestry, mining and commercial agriculture on sacred places and indigenous people in Venda.

The authors also conducted field visits to some of the sacred sites to document potential and real threats to cultural biodiversity and evaluate their meaning to the Venda in the context of sustainable development. One of the authors (MM) is indigenous to the area and was invited to participate in a reburial ceremony involving a visit into the heart

of the sacred Thathe forest. This ceremony is highly sanctioned by the Tshidzivhe Clan, who do not allow video recording and photography of the rituals as they proceed deep into the forest. He was also not allowed to observe some ritual practices, such as those concerning supernatural characters. This was due to related taboos associated with the forest. However, by observing some of the actions of the clan closely, it was possible to learn more about relevant cultural activities associated with sacred sites and habitats, as well as the amount of respect custodians and villagers have towards them. This also presented the researchers with deep insights into why intangible elements are critical towards tangible cultural heritage conservation and the well-being of indigenous peoples.

The Venda worldview derives from their mythology and mythography [29]. To understand cultural landscapes and how some indigenous communities interact with these, we highlight how legends and myths are critical to biodiversity preservation and to attaining sustainable environments. Venda is rich with legends and myths central to the heritage of indigenous people. Legends and myths are not merely 'stories', but rather, a reality accounting for the formation and evolution of most of the objects that made-up their landscape and that continue to shape their identity. Since they influence and configure Venda worldview, they are central to their sustainability.

In the process of evaluating Venda mythology and mythography, we discovered crucial gaps in knowledge, arising from our misreading and misunderstanding of core social sciences such as anthropology, archaeology and history [18–23]. Intangible heritage is mostly located in narratives told through myths and legends and these are often missing in landscape studies. Myths and legends are inscribed within specific landscapes. Such landscapes are the physical illustration of how people have related to and transformed their environments [29,30] in ways meaningful to them, how they manipulated and influenced change, impacting on themselves and the environment. Therefore, the relationship between humans and landscape is an everlasting one and the foundation of all other relations in human society [30,31]. Sustainability is located within such a relationship.

Indigenous forests provide a plethora of basic needs which are either consumptive or non-consumptive. Indigenous forests of Venda have sacred sites which have always been protected by particular clans from time immemorial [32–35]. These clans are entrusted with traditional secrets and folklore related to the continuation of ancient rituals and ceremonies. Sacred sites are therefore integral to the people of Venda such that almost every village has a sacred site. We conceptualise these in the broader context of cultural heritage biodiversity and sustainable development.

## 3. Conceptual Framework

Current discussions on biodiversity-related environmental change point to what has been referred to as the Holocene extinction [33,34,36], and these are informed by human accelerated deterioration of terrestrial and marine environments from the 1950s onwards. Ultimately, the damage incurred on the ecosystems undermine targets that have been set by the United Nations, especially the Sustainable Development Goals (SDGs) which are aimed towards addressing hunger, poverty, health, water, urbanisation, climate change and the environment [1,2]. While conservation of biodiversity including cultural heritage is vital [30], there is need to acknowledge local ways of knowing and indigenous worldviews. Cultural heritage has so far received limited treatment when it comes to sustainable development [30].

It is evident that globalization, urbanization and climate change are threatening cultural heritage and weakening cultural diversity. To address this, measures to promote the safeguarding of cultural heritage within the global development agenda, where concrete actions are needed to integrate cultural heritage conservation with the sustainable development goals, are required. We, however, lament the absence of cultural heritage in the formulation of Sustainable Development Goals and the targets that these goals seek to meet. We also encourage mainstreaming cultural heritage into this agenda, which includes biodiversity. Sustainable Development Goal 15 (SDG 15), for example, seeks to

"Protect, restore and promote sustainable use of terrestrial ecosystems, sustainably manage forests, combat desertification, and halt and reverse land degradation and halt biodiversity loss" [1]. This is achieved by promoting efforts to end deforestation and restore degraded forests by promoting the implementation of sustainable management of all types of forests, halt deforestation, restore degraded forests and substantially increase afforestation and reforestation globally. The same applies to ending desertification by 2030 by restoring degraded land and soil. To help achieve this, local and national planning is a requirement for state parties, and for the former, local and indigenous communities are fundamental. Although only one goal has been singled out here to highlight aspects of biodiversity, Sustainable Development Goals are integrated and indivisible. SDG 15 links with the Aichi Biodiversity targets and strategic plan (2021–2030) that seek to address the underlying causes of biodiversity loss by mainstreaming biodiversity across government and society, reduce the direct pressures on biodiversity and promote sustainable use, improve the status of biodiversity by safeguarding ecosystems, species and genetic diversity, enhance the benefits to all from biodiversity and ecosystem services and enhance implementation through participatory planning, knowledge management and capacity building [31]. We also acknowledge the African Union Agenda 2063, which ascribes to these values.

According to the Intergovernmental Science-Policy Platform on Biodiversity and Ecosystem Services report [33,34], biodiversity losses from disturbances caused by humans are severe and continuous. We show how this is happening in Venda in and around three sacred sites and landscapes.

## 4. Results

There are three major scared sites or landscapes central to Venda worldview, namely Thathe forest, Lake Fundudzi and Phiphidi waterfalls (Figure 2). According to legend, half-humans collected water from the waterfalls, travelled into the adjacent forest for rituals, and finally to the lake for rain petitioning. This cultural landscape is under serious threat due to mining and agriculture. Significant colonial and corporate developments in mining and agriculture, have been touted as crucial investments that would bring employment among local and indigenous people and help reduce poverty and hunger. On the contrary, these developments are biodiversity loss, increasing hunger and poverty. Such biodiversity losses to Venda worldview and identity are a threat to Sustainable Development Goal 15 (Life on Land). We illustrate this with regards to the three sacred sites and landscapes.

### *4.1. Thathe Forest*
#### 4.1.1. Biodiversity

Thathe sacred forest is one of the few remaining, largely undisturbed moist forests in southern Africa [32] (Figure 3). The original extent of the sacred forest before commercialisation is unknown. Surrounded by the villages of Vondo, Tshidzivhe and Tshilungwi, and commercial timber plantations, what remains of this sacred forest is about 3807 hectares [32]. Thathe is a botanical reserve that falls within the northern mist belt, receiving annual rainfall of approximately 1500 mm [32]. The vegetation found therein is of subtropical provenience. Its tall and diverse trees are multi-layered, with a well-developed canopy (Figure 3).

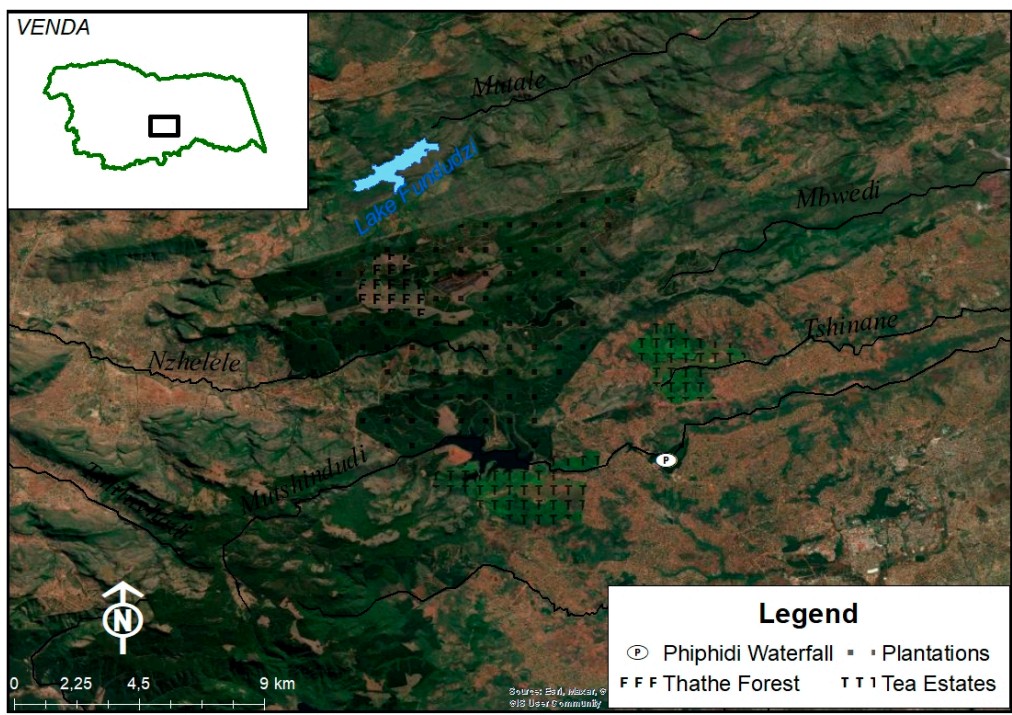

**Figure 2.** The region of Venda showing the location of Thathe forest (F), surrounded by pine and eucalyptus plantations, Lake Fundudzi, the tea estates and Phiphidi waterfalls (P) (Map, prepared by Brenda Makanza).

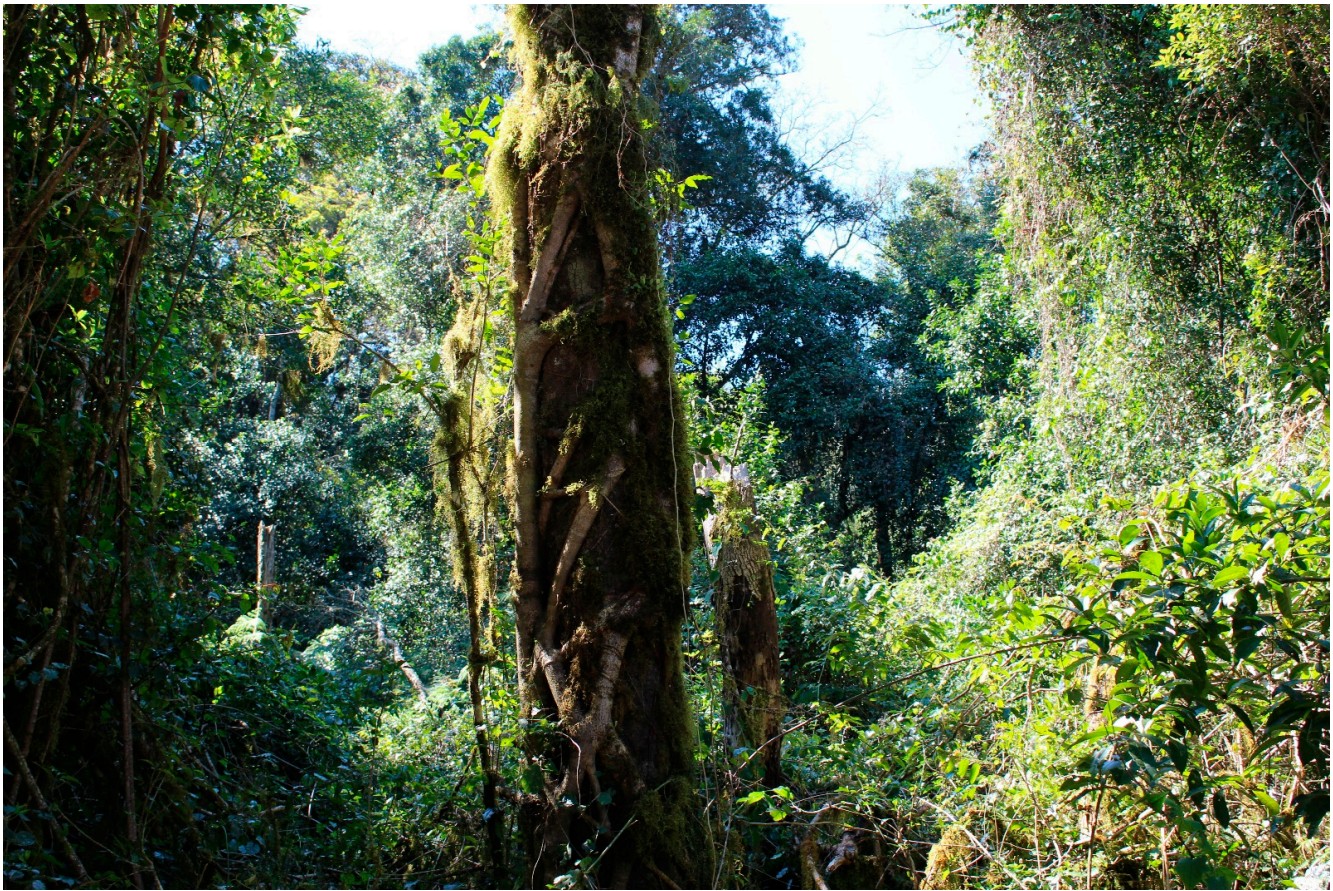

**Figure 3.** Thathe forest, showing the giant hardwoods, ferns, undergrowth and creepers (Image, by Munyadziwa Magoma).

Thathe flora is characterised by giant hardwoods such as yellowwoods, ferns, thick undergrowth and creepers. *Cryptocarya liebertiana* (tropical quince) is the most common. Other conspicuous trees include *Drypetes gerrardii* (ironwood), *Rawsonia lucida* (forest peach), *Combretum kraussii* (bushwillow), *Ficus natalensis* (Natal fig), *Podocarpus latifolius* (yellowood), *Xymalos monospora* (lemonwood), *Cussonia spicata* (cabbage tree) and *Oxyanthus gerrardii* (Whipstick Falseloquat). The forest is impenetrable and dense, attracting several bird species, including the chorister robin-chat (*Cossypha dichroa*), the white-starred robin (*Pogonocichla stellate*), the Knysna turaco (*Tauraco corythaix*), the yellow-streaked greenbul (*Phyllastrephus flavostriatus*) and the orange ground-thrush (*Geokichla gurneyi*) [32]. It is how the local Venda people relate to the forest that make this biodiversity unique.

### 4.1.2. Cultural Significance

The name 'Thathe' derives from the Venda word 'thatha', meaning to "chase away". According to Venda, a lion lived in this forest. Since the lion used to chase villagers herding cattle and collecting firewood in the forest, it was named Nethathe, meaning "the chaser", or "the owner of Thathe". This must have been a reincarnation of a Venda chief by the same name. Thathe is also the place where the sacred python or the python god that resides in the nearby Lake Fundudzi disburses some of its time (personal communication with Chief Netshiavha). This sacred forest is also guarded by a lightning bird, Ndadzi, and is home to the half-spirit, half-human, predatory and apparently clumsy people. Although the forest is rich in firewood, no person dares to collect wood for fear of being traumatised by spirits residing therein. It is also believed that if one fetches wood from Thathe, such wood can transmute into a harmful snake [37].

Thathe forest is used by the Netshidzivhe family to bury or rebury their chiefs and to venerate their ancestors. According to Venda traditions and our conversations with Chief Tshidzivhe, the late Chief Nethathe instructed his people that to be buried in the cave where he stored his herbs and charms. He also ordered that all children and future generations of Thathe, whether male or female, be buried in the rainforest, while the remains of chiefs were to be placed in a cave. To this day married women whose surnames are Nethathe, Netshidzivhe or Netshitangani are buried within Thathe. It is also believed that the forest is guarded by the spirit of Khosi Nethathe, who reincarnates as a white lion. This spirit lion also helps bring rain, but such rain may not fall if trees in the forest are cut. Nethathe himself was a healer, diviner, herbalist and magician, and this is why legend says he was able to shapelift (in therianthropic terms) himself into a white lion.

### 4.1.3. Appropriation

Thathe has been used for commercial forestry since 1945 [38]. The original extent of the sacred forest before commercialisation remains unknown. Based our conversations with Chief Netshidzivhe, who took us through parts of the landscape where they conduct rituals and ceremonies for the reburials of chiefs, we give an estimate of 40,000–50,000 hectares. Commercialisation resulted in pine timber plantations (Figure 4), which interfered with the common property tenurial system customary to the custodians of Thathe. They resisted, setting fire on plantations and uprooting pine seedlings. These acts of resistance were interpreted as anger by the spirits of Thathe, avenging the destruction of the sacred forest. Despite pleas from headman Netshidzivhe in 1949 to preserve parts of the forest, the South African government continued with the appropriation [37,38], such that the preserved sacred forest is now entirely surrounded by pine trees.

Another development undermining the cultural as well as environmental integrity of Thathe forest is mining. Unknown to local communities, a mining company, the Mammba Metal Group, has been prospecting the forest for precious minerals such as coal, diamonds and gold since 2018, and, apparently, with the approval of the South African government. It is suspected the mining company managed to prospect without proper environmental impact assessments. Our informants, who chose to be anonymous, expressed concern about the proximity of the mining claims to the forest since what remains of this forest is

not sufficiently buffered from such high impact developments. The prospecting is already impacting negatively on the nearby sacred Lake Fundudzi, endangering indigenous plants endemic to the area and polluting water resources. The prospecting is also likely to reduce river water supply and desiccate the sacred lake. These developments are triggering water scarcity, in a region already ravaged by droughts and dry periods. Many people in Venda are aware of the devastating effects of the open-cast Tshikondeni coal mine, some 120 km to the north-east of Thathe (Figure 5), a development originally touted as generating employment and ending poverty.

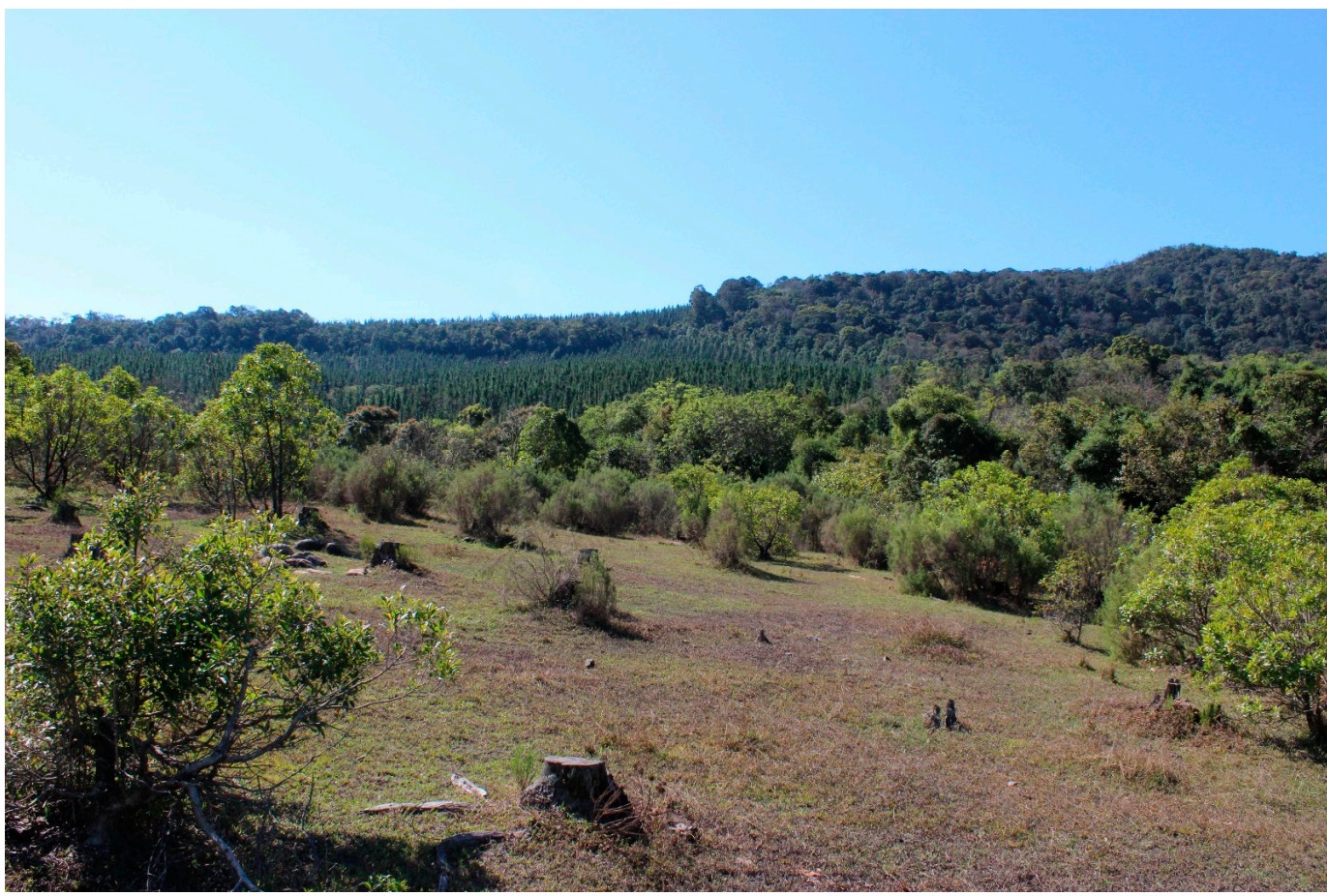

**Figure 4.** Commercial plantations around Thathe forest. Note the scars timber harvesting is inflicting on the environment (Image, by Munyadziwa Magoma).

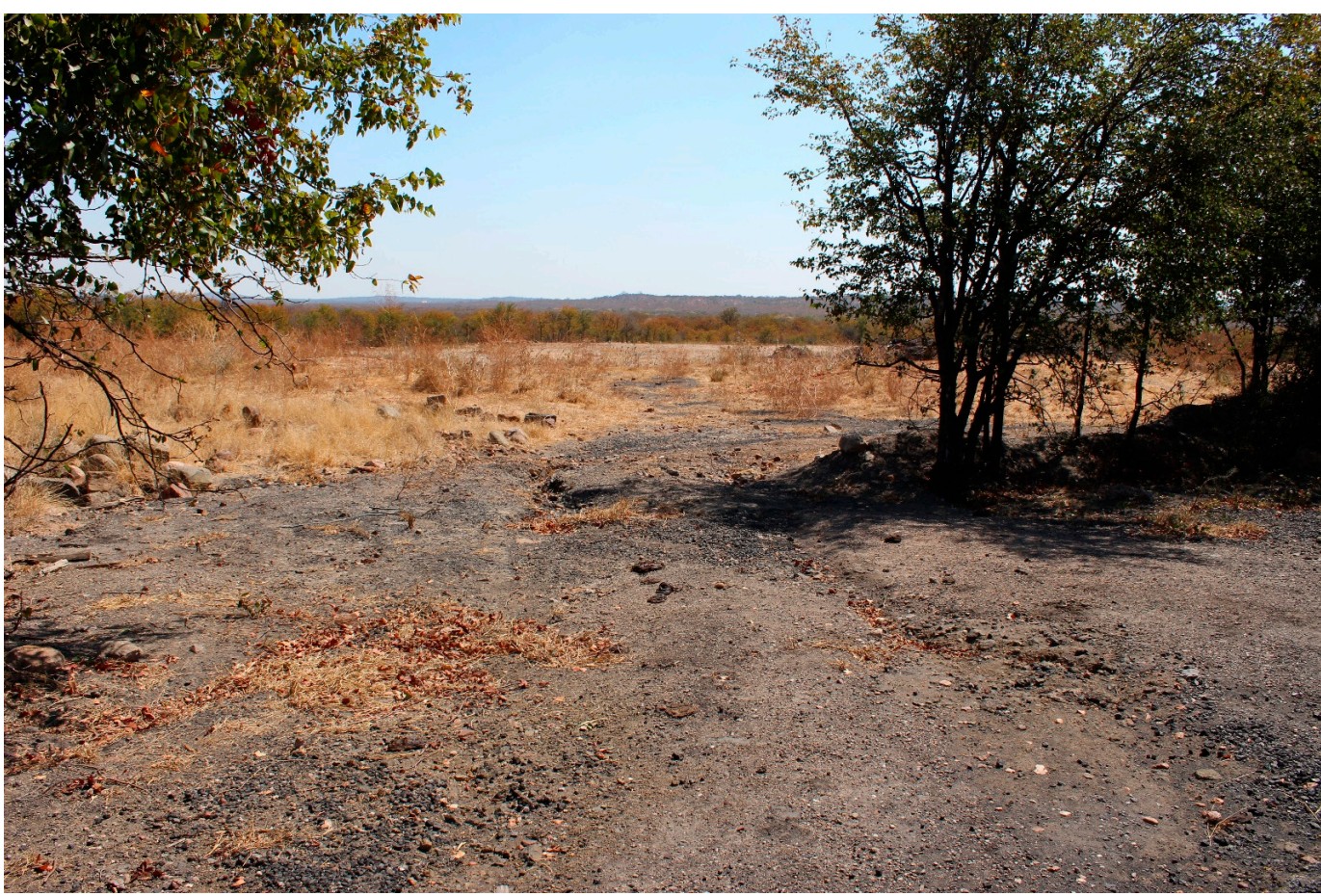

**Figure 5.** The destructive effects of open cast mining in Venda, following coal mining at Tshikondeni. The discovery of minerals such as coal in much of Venda has led to the destruction of not only the physical environment, but also people's livelihoods (Image, by Munyadziwa Magoma).

### 4.2. Phiphidi Waterfalls

### 4.2.1. Cultural Significance

Phiphidi waterfalls (Figure 6) are located about 8 km from Lake Fundudzi. Its traditional custodians are the Tavhatsindi of the Ramunangi Clan, who claim to have originated from there [39]. The landscape of Phiphidi, including the river, the falls and surrounding forest, is considered sacred. Venda elders informed us that the part considered most sacred is the rock above the plunge, locally known as Lanwadzongolo and the pool below the waterfall, referred to as Guvhukuvhu. Complex laws and rituals govern behaviour to be observed at the waterfalls by everyone. It is believed that the area is inhabited by ancestral water spirits, which are appeased annually through ritual offerings. There are reports of '*malombo*' or consecrated musical performance reserved for ancestral rites, coming from Phiphidi waterfalls [40].

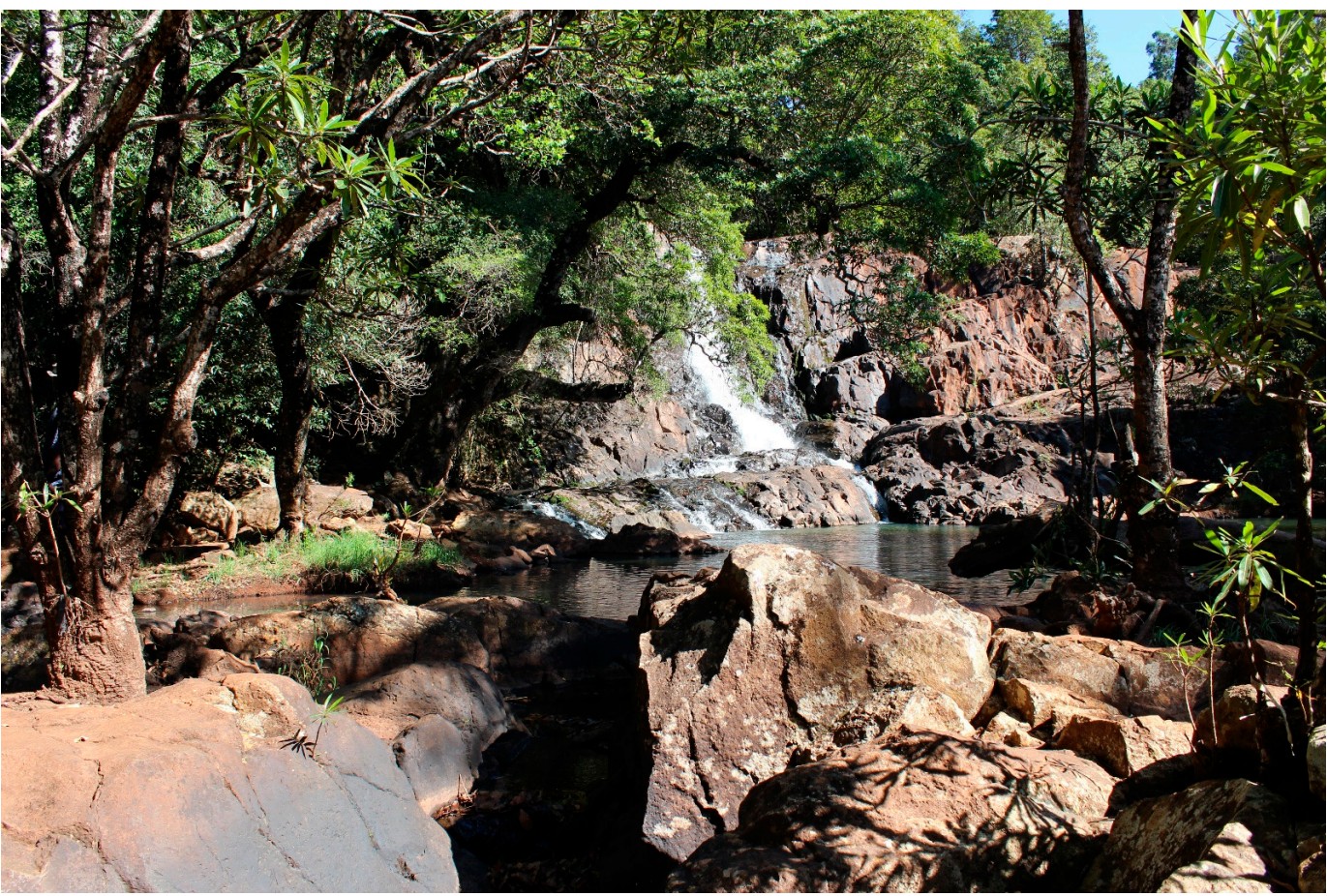

**Figure 6.** Phiphidi waterfalls and associated forest (Image, by Munyadziwa Magoma).

### 4.2.2. Appropriation

When the area was proclaimed as a separated administrative region under the South African political system of racial segregation (apartheid) in 1979, smaller clans, including the Ramunangi, were silenced and some sacred sites were destroyed [39]. Phiphidi waterfalls were affected, when the government opened it up for tourism. This saw the construction of roads, pathways, picnic sites and ablution facilities for tourists, on culturally sensitive land and *inter alia*, in breach of local taboos and cultural rules. These falls are central to the Ramunangi's relationship with ancestral spirits [40]. However, their ownership of the place is not recognized, limiting the Ramunangi's ability to protect Phiphidi from tourism development. The Ramunangi are still denied full access to the waterfalls for ritual and ceremonial performance. Lanwadzongolo—one of the site's most sacred areas—was recently destroyed as part of a road-building project, irretrievably defacing the surrounding sacred forest. The clan is now seeking legal redress to regain full custodial recognition and access to the site [39].

Agricultural developments (Figure 7) nearby Phiphidi waterfalls and forest have further resulted in land clearance, which in turn have eroded and degraded more sacred places. In 1973, Chief Tshivhase made available land for tea estates on the banks of the Mutshindudi River [41]. Although these estates were so productive that many assumed Venda might become the 'Ceylon of Southern Africa [42], this was attained at the expense of sacred places including burial sites that were desecrated in the process. Indigenous people were relocated to make way for the growing of tea. The tea estates presently cover some 584 hectares and within it are sacred sites and burials, which although spared from destruction, remain isolated since their custodians and communities have been relocated.

### 4.3. Lake Fundudzi

4.3.1. Biodiversity

Immediately below and adjacent to Thathe forest is a sacred waterscape—Lake Fundudzi (Figure 8). This is a landslide dam, approximately 3 km long and 1 km wide, covering an area of about 144 hectares, with a maximum depth of 27 metres. Geologists say the lake was formed some 10,000 years ago. The Mutale River channel enters the lake through the middle of its bed, running in a south-west to north-eastern direction. The lake is part of the biodiversity which includes the surrounding mountainous area, where Thathe sacred forest is located [43,44].

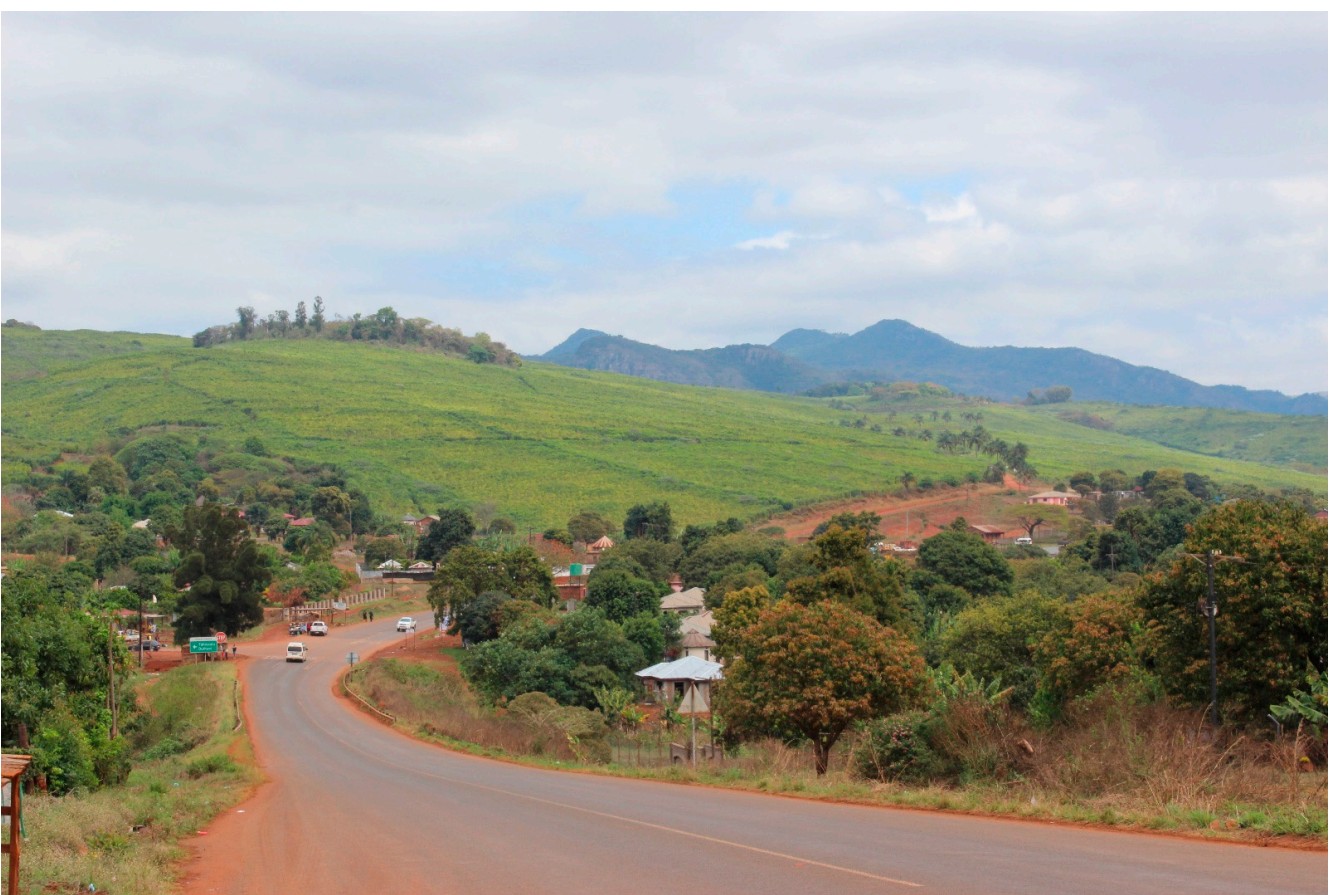

**Figure 7.** A view of some of the tea estates in Venda. The background shows a groove located in the middle of the estates, which is sacred site (Image, by Munyadziwa Magoma).

4.3.2. Cultural Significance

Lake Fundudzi's unique geomorphic formation is captured in Venda mythology. According to one Venda narrative, Lake Fundudzi was formed when a leper passing through a village located on the banks of Mutale River was denied food and shelter. In displeasure, the leper cursed the villagers which triggered landslides that caused the river to flood. Villagers drowned and a lake was created in the process [45]. Villagers who drowned in the lake became half-human people and it is still believed they still reside there. Villagers living close to the lake claim hearing screams and bellowing of people and cattle, reminiscent of drowning [46]. The custodian of Lake Fundudzi claims sighting half-human people in the lake. During rain-petitioning rituals, these half-humans are said to move out of Lake Fundudzi to fetch water from Phiphidi waterfalls, and bring it to the lake. This water then evaporates and generates rainfall.

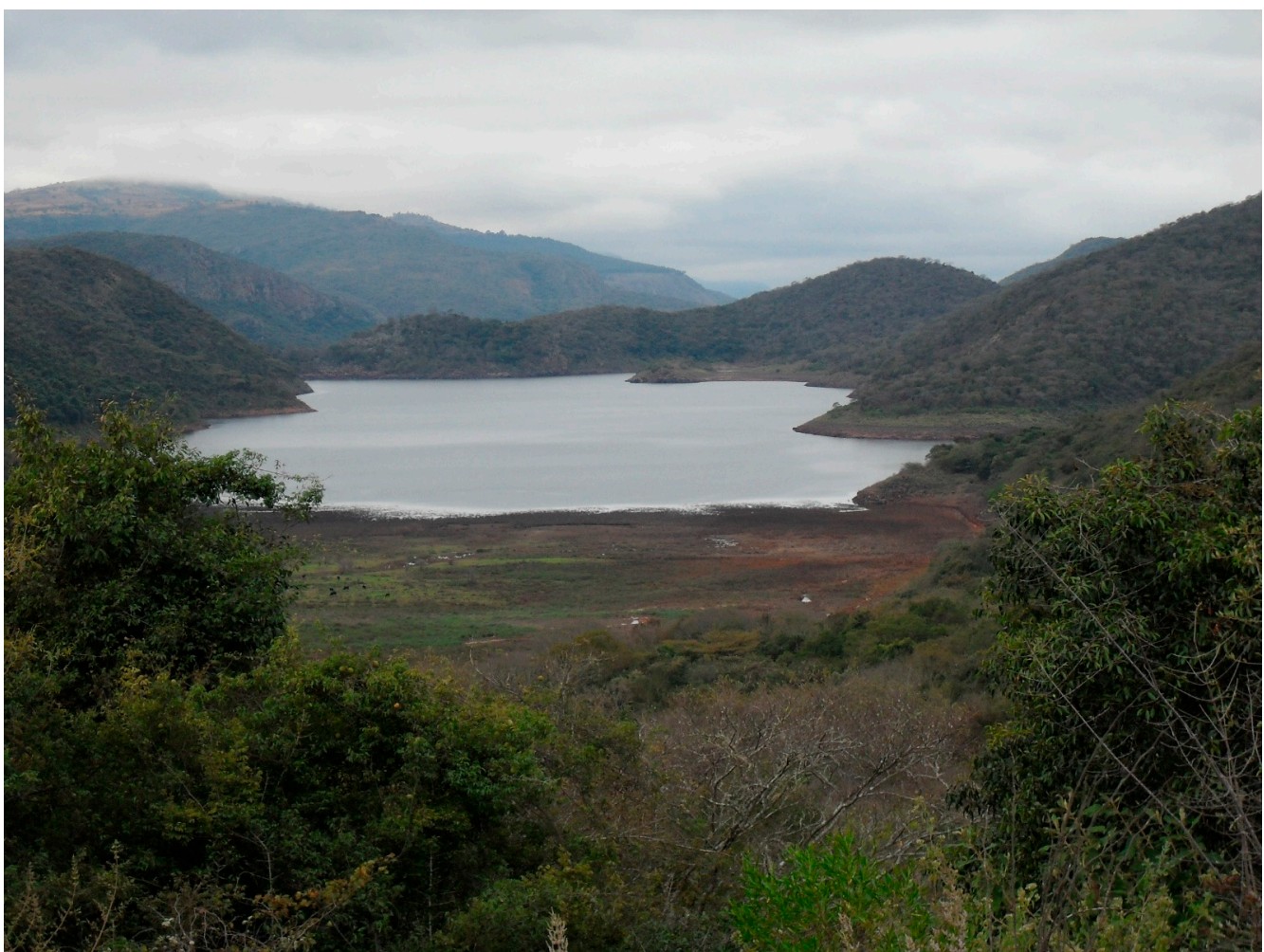

**Figure 8.** Lake Fundudzi. The foreground shows considerable sedimentation, as result of ecological degradation of the lake's catchment, accelerated by commercial plantations, mining and human settlement (Image, by Innocent Pikirayi).

Central to the mythology around Lake Fundudzi is the white python spirit of fertility that resides there [45]. In ancient times, this python was said to copulate with women from nearby villages, at night while they were asleep. When caught in the act by a woman who rebuked it, the snake was upset, fled and hid deep in the lake. This triggered a terrible drought, which only ended when the woman waded into the lake to copulate with the python. To prevent subsequent droughts, young maidens were sacrificed yearly in the lake, to satisfy the python's lust. The Venda girls' initiation ceremony, *domba*, is performed annually at the lake is in commemoration of the python god [47]. It is also believed a white crocodile lives in the lake. When Venda kings died and their remains were placed in the lake, this crocodile coughs up a stone, which the new king had to swallow. The spirits of the ancestors of the Venda people are believed to reside beneath Lake Fundudzi and are guarded by this crocodile. These and other rituals and ceremonies are also performed around the lake not only to honour the python and the crocodile, but also to ensure continued fertility and peace in the land [45].

### 4.3.3. Threats

There are major threats to Lake Fundudzi if developments within its broader landscape are considered. Since the 1990s, the catchment of Lake Fundudzi has been subjected to afforestation though gum-tree (Eucalyptus) plantations, mineral prospecting, human settlement and dry land cropping (see Figure 8, above) [46]. These activities have resulted in forests continuously replaced by croplands and orchards, placing considerable pressure

on the catchment. In addition, some hillsides have been cleared to prevent wildfires from encroaching eucalyptus plantations [46].

Lake Fundudzi is experiencing rapid sedimentation from its catchment due to increased human settlement and agricultural activity (Figure 9). Local communities are also using portions of the lake inundated during the rainy season for vegetable production. We observed how nearby settlements are developing fruit orchards, further generating erosion and silting, which may result in the disappearance of the lake.

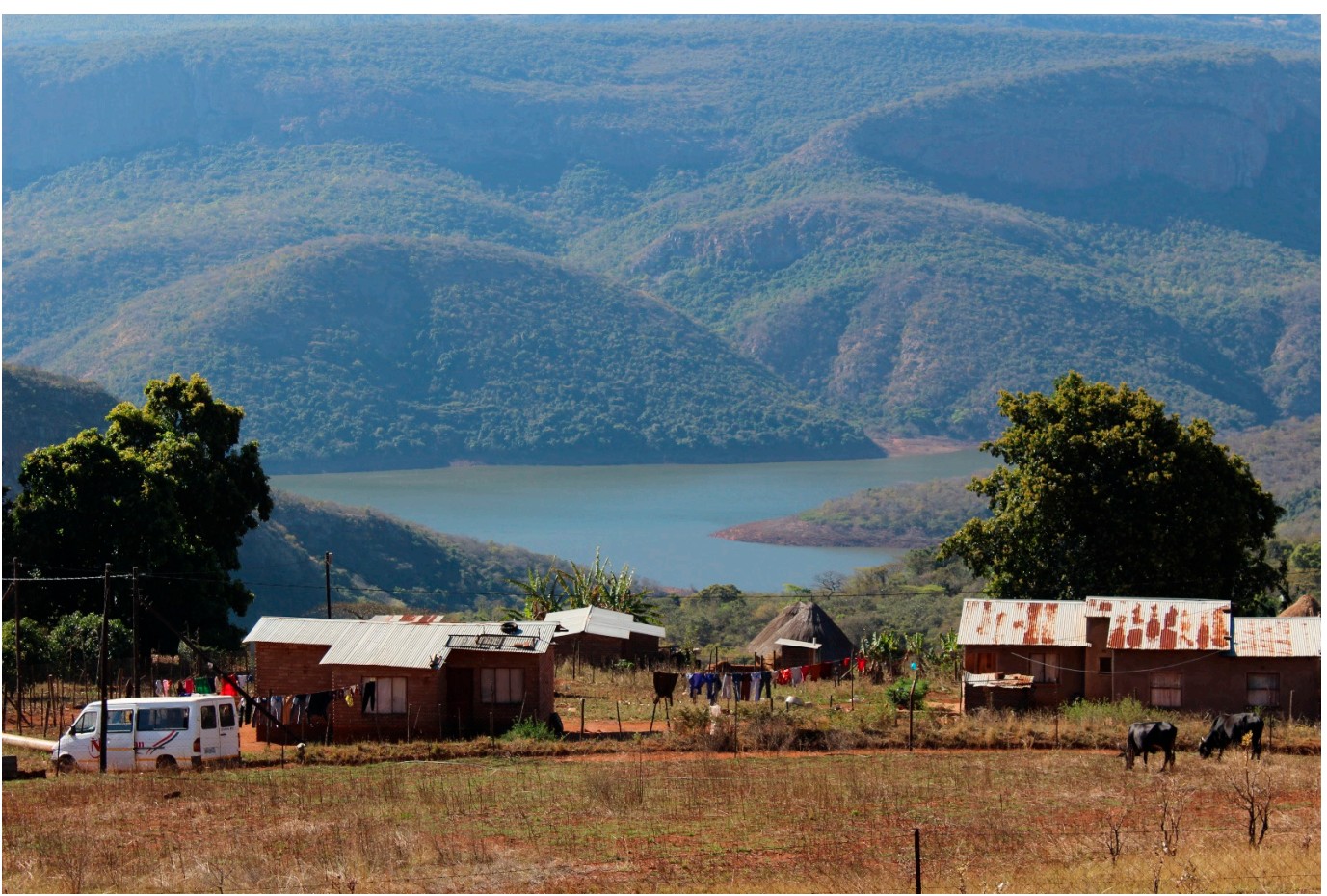

**Figure 9.** Human settlements encroaching Lake Fundudzi (Image, by Munyadziwa Magoma).

## 5. Discussion

In Venda, human-driven biodiversity loss occurs concurrently with loss of forests, destruction of waterscapes and degradation of other ecosystems. Forests are rapidly disappearing and waterbodies silting due to increased human settlement, farming, mineral prospecting and infrastructural development. The destruction of existing natural and cultural habitats results in local and indigenous communities losing food resources and living spaces. What is central about Phiphidi waterfalls, Thathe forest and Lake Fundudzi is their sacredness. Destroying and desecrating these sacred sites and places will result in famine and hunger. Thathe forest is a regenerative 'spring' for Venda farmlands including areas east of the Soutpansberg and as such speaks to sustainability of livelihoods among the local and indigenous peoples. Thathe forest is a source of a number of rivers, and though miniscule, may be equated to the Congo and Amazon rainforests, which act as 'lungs of the planet', drawing critical carbon dioxide and oxygen in and out, respectively [48]. As such, Thathe, should be seen as the 'lungs of Venda', which are vulnerable to modern developments.

What we found interesting in our study is how integrated the three sacred sites and places are within the landscape as well as to the lives and origins of the Venda. How people make sense of these sites and places, e.g., the formation of the Lake Fundudzi, local human experiences and understandings of environmental catastrophes such as the devastating effects of landslides, loss of soil fertility and flooding, etc., are credible understandings that speak to modern science. In this regard, myths and legends are vital towards biodiversity, including cultural heritage conservation [49,50].

There are possible solutions to human-driven biodiversity loss in Venda and how these may help towards the attainment of SDGs. An obvious approach would be to call upon the assistance of conservationists to assist with documentation, environmental monitoring and preservation of the forest, the waterfall and the lake. This would protect the forest from unnecessary exploitation, while disincentivizing behaviour that contributes to habitat loss and degradation [51,52]. This, in our view, has not been done. Sustainable development involves planning to promote growth while preserving environmental quality, which central in areas, for example, affected by intensive agriculture and large-scale mining operations. This must be considered when creating new farmland and human settlements. Current developments within and on the precincts of Thathe forest, Lake Fundudzi and Phiphidi waterfalls do not speak to sustainability—given the negative effects of such developments elsewhere in Venda such as the recently closed coal mine at Tshikondeni. There is also no recourse to traditional knowledge systems in safeguarding sacred landscapes such as the forest and the lake.

It is evident, from our interactions with these landscapes that the sacredness of Phiphidi, Thathe and Lake Fundudzi developed as a corpus of Venda worldview upon realising how vital these were to the well-being of the indigenous people. Such landscapes are localised ecosystems, which also regulate human societies. Interfering with their operations undermines indigenous worldviews irretrievably. What is happening in Venda as far as such landscapes are concerned is a macrocosm of how the world is losing tropical rainforests through unsustainable human exploitation.

Laws protecting the destruction of cultural heritage must be enforced. Lake Fundudzi and Thathe forest have been placed under protection since 2014, through the Heritage Resources Act, 25 of 1999. Despite this, there has not been systematic monitoring of these sites and landscape to assess the negative impacts triggered by large-scale mining, commercial forestry and large-scale farming. While myths and legends are central in conserving sacred sites and landscapes [53–55], they are, however, rarely given prominence when it comes to cultural and biodiversity conservation and sustainable development. In this paper we demonstrate how vital this is for Venda cultural heritage, their identity, well-being and sense of place.

Rising population also creates an economic demand, which explains why areas like Venda are under serious threat from mining and agriculture. According to the South African National Environmental Management Act, it is a serious offence to commence environmental impact assessment without authorisation. However, existing laws and policies must help promote sustainability. Venda traditional leadership seeks to conserve biological diversity of the landscape through comprehensive ecosystem management and through equitable sharing of the benefits of tourism between local and indigenous communities. While this is viewed as a way of reducing poverty among these communities, it requires their consent and full participation in the planning and management of development projects. This is evident in the safeguards local and indigenous communities were offered by the Interim Protection of Informal Land Rights Act of 1996, ratified by two court judgement in 2018. In both judgements, although traditional leadership had sided with developers without consulting their communities, the courts ruled that "informed consent" of entire communities was paramount. It is regrettable that traditional leaders in Venda are not able to stop such developments because some of them have openly sided with developers.

Sacred landscapes require holistic approaches when it comes to their conservation [54,55]. These landscapes are intimate spaces, susceptible to cultural erosion if the focus is on tangible aspects of such heritage [8]. Much less overlooked is the intangible especially when it comes to cultural heritage conservation and sustainable development. Further, Western conventional heritage conservation understandings are generally inappropriate and unsustainable when preserving such places. There are unquantifiable connections between people and place that, if not considered, may result in memory erasure and the un-inheriting of heritage places. The role of mythology as cultural identity and as heritage is known [54,55]. With regards to Thathe and Lake Fundudzi, the use of myths and legends is central to appreciate how and why indigenous people have conserved these sustainably [29,55]. We conclude that myths and legends of the Venda have cultural sanctions and prohibitions whose values serve to conserve sacred places, which in turn are the abode of their ancestors and the core of the worldview.

## 6. Conclusions

Despite threats to its human-driven biodiversity, Venda's autochthonous people find considerable meaning and power in their heritage, both natural and cultural. Their landscape is replete with considerable social memory, which anchors individual people, groups and communities and their memories to broader, societal understandings of their past. Venda culture is embedded in an environment with its own histories and sense of the past. Our paper has shown how in such a world characterized by disparate approaches to development, with negative impact on indigenous communities, knowledge of local histories, myths and legends is imperative. Already, with the threats to the forests and waterscapes located in the heart of Venda, it is highly unlikely that some sustainable developments aligned to the protection of heritage, will be attained.

We reiterate that Sustainable Development Goals (SDGs) are about the future, and how any generation perceives such [1–3]. If we recall Pierre Teilhard de Chardin (1881–1955) [4,5] we ask whether modern developments in Venda are in a position to give the present and next generations reason for hope. This may not be so, given the human-driven biodiversity losses incurred due to environmental degradation, causing hunger and poverty in the process. Though the environmental circumstances have radically changed since the time of Homo erectus *pekinensis*, we wonder what gave hominins reasons for hope, with the universe constantly shifting towards a state of greater complexity [6]. What Teilhard de Chardin identified in this shift towards greater complexity was the development of an ecosystem that was primarily human-driven. What he was not able to predict then, was how potentially destructive such ecosystems would be and the environmental crisis confronting humanity today.

We do not understand how human-nature interactions over time have created premises on which the environment is conserved to serve not only the present but also future generations. This is key to realise most if not all the sustainable development goals. In Venda, there are consequences when the values ascribed to sacred landscapes are ignored, in preference for developments such as mining, commercial forestry and large-scale farming.

**Author Contributions:** Conceptualization, I.P. and M.M.; methodology, I.P. and M.M.; formal analysis, I.P. and M.M.; investigation, I.P. and M.M.; resources, University of Pretoria Library.; data curation, University of Pretoria Repository; writing—original draft preparation, I.P. and M.M.; writing—review and editing, I.P. and M.M.; project administration, M.M..; funding acquisition. All authors have read and agreed to the published version of the manuscript.

**Funding:** This research received no external funding. The APC was funded by MDPI, who offered 100% discount to papers submitted to the special issue "*Heritage as a Driver of the Sustainable Development Goals*" before 31 August 2021 (Author voucher discount code: eebf9276deda13f6).

**Institutional Review Board Statement:** This study was conducted within the framework of the project "Lake Fundudzi: Water, Landscape and Cosmology of Thathe-Vondo and Makonde Cave, northern South Africa". It did not involve humans or animals. Permission for the research was

granted by the University of Pretoria Faculty of Humanities Ethics Committee (Reference No.: 1131 1828 [HUM046/1120]), approved on 31 May 2021.

**Informed Consent Statement:** Not applicable.

**Data Availability Statement:** All the data and accompanying metadata generated from the project will be stored and managed by the University of Pretoria Research Data Management System, which is an accredited data repository (https://researchdata.up.ac.za/). The data will be embargoed for at least two years to allow researchers and postgraduate students to publish research proceedings from the raw data generated from the project.

**Acknowledgments:** This is a scoping paper on re-imagining heritage in Africa, examining it from the context of cultural losses inflicted by commercial mining, agriculture and other modern infrastructural developments. The authors would like to thank the elders as well as the ancestors of Venda for challenging them to think differently, and in the process, appreciate better understanding of their origins, histories and worldview. We thank in particular Ndinae Ramasiwela for assisting with the surveys and the late Ntsandeni Netshiavha of Lake Fundudzi for sharing his wisdom with us.

**Conflicts of Interest:** The authors declare no conflict of interest.

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
