# Peer review of "Retrieving Intangibility, Stemming Biodiversity Loss: The Case of Sacred Places in Venda, Northern South Africa"

_heritage, doi:10.3390/heritage4040249_

Round 1
Reviewer 1 Report
- Topic of this paper is important and appropriate for Heritage journal. There is undoubted problem about people Venda and negative consequences of globalization.
- Despite authors wanted to point out clearly what are the main problems of this paper, content of the paper lacks basic scientific paper structure: hypothesis, methodology, discussion on given results. It is very hard to understand content of the cultural landscape based just on myths and legends.
- There is poor official or unofficial data about people, environment or processes mentioned.

Author Response
Response to Comments from Reviewer #1:
The reviewer doubts the value of myths and legends and their efficacy in approaching landscape studies and in this context, is critical towards the conservation of these meeting sustainable development goals. However, this is what Western scholarship in literature studies has been about, this at the expense of literature studies from the Global South, where myths and legends are not regarded as valuable sources of history and as critical in landscape studies. We differ with his/her views.
However, we do accept his/her comments regarding the structure of the paper, and we have addressed that starting with the abstract, the introduction, the conceptual frame (including the research questions/hypotheses), the data, results and conclusions.
The comment to the effect that “There is poor official or unofficial data about people, environment or processes mentioned”, is unclear to us. Perhaps it is a matter of phraseology, as the reviewer admits she/he is not qualified to comment on the English. The data that this paper uses/employs is found in the public domain, in both published and unpublished sources.
With regards to other areas in the paper that need improving (based on responses in the reviewer rubric), we have paid attention to the content, the arguments and discussion, as well as the references, which are now presented using the journal style and format.
Please ALSO see attachment, which consolidates the comments from both reviwers #1 and #2.

Reviewer 2 Report
Dear authors,
First of all, I have to say that the article is very interesting for me. People living outside of Africa quite rarely receive information on cultural heritage of such communities. Especially the part of myths and legends of Venda people is really interesting.
Following the basic requirements for writting of scientific paper, I have to come up with several important comments.:
Abstract: Please re-write the abstract in a way that a reader can clerly follow the aim of your research, methodology used and results you have achieved.
Introduction and Research context: some parts of research context should go into the introduction and other parts should be re-written in a way that we can follow the methodology of your research. I personally would like to recommend to put down the introduction with 4-5 paragraphs on sustainability goals, biodiversity and culture heritage as a theoretical background and then 2-3 paragraphs on Venda problems. The last paragraph should outline the aim of your research.
Methodology: Methodology of your research is completely missing.
Results: Whar are results of your research? Very unclear? Is it the description of biodiversity loss in Venda? this should accompanied by some figures, statistics, etc.....
Discussion and conclusions: usually discussion and results correspond with results. I can not see this relation in your article
Other comments:
152 – 187 lines are not necessary, delete them
Please follow the template of Heritage mdpi journal (references in the text, etc....)!!!!!!!!
Author Response
Response to Comments from Reviewer #2:
We appreciate the detailed comments of Reviewer #2 especially as he/she sees value in myths and legends from the perspective of cultural heritage.
We have addressed the comments from the reviewer as follows:
Abstract: We have changed the abstract to reflect the content and structure of the paper as suggested.
Introduction and Research context: The reviewer recommends that some parts of research context should go into the introduction and other parts should be re-written in a way that readers can follow the methodology of the research. He/she further recommends writing the introduction with 4-5 paragraphs on sustainability goals, biodiversity and cultural heritage as a theoretical background and then 2-3 paragraphs on Venda problems. The last paragraph should outline the aim of the research. These are highly constructive suggestions, which we have attempted as much as possible in our revised version.
Methodology: The reviewer also says the methodology of the research is completely missing. We agree, as this project/paper is still at the conceptual stage. In response, we have now inserted the methodology section in the paper, reflecting some desktop work, as well as preliminary fieldwork where we informally interviewed traditional leaders, developers and local communities within Thathe and adjacent areas, which is the research area.
Results: The reviewer asks about the results of our research, which she/he says are unclear. He/she wonders whether these are merely the description of biodiversity loss in Venda, and if so, would like that accompanied by some figures, statistics, etc. We have clarified the results, which point to the loss of sacred sites, which are critical towards biodiversity conservation in the research area. Based on the qualitative approaches employed, there are no statistics to support this, but rather qualitative indicators in terms of what the Venda communities within Thathe are experiencing.
Discussion and Conclusions: The reviewer states that usually discussion and results correspond with research results, which he/she did not see in our article. In response, we have aligned the results with the discussion and conclusions.
Other comments raised by the reviewer:
The reviewer recommends we should delete lines 152 – 187 as these are not necessary: After some thought and debate, we have seen the logic behind the recommendation, though our original intention was to highlight how myths and legends are valorized in a Western context, but despised in an African context, and not seen in any way as important towards understanding cultural landscapes and biodiversity. We have deleted the lines, but retained some of the references, which we have re-assigned to a number of sentences in the paper highlighting the same issues.
We will follow the template of Heritage MDPI journal, improve the content, arguments and discussion, etc., as already highlighted by Reviewer #1.
(Please ALSO see attachment)

Round 2
Reviewer 1 Report
I don't see any changes in this revised version. Since authors ignored my comments in the paper I don't see reason to change previous (First decision) on it.
-------------------------------------------------
After I review the manuscript once again, it seems to me the paper is ready for publication. Authors did almost all the changes needed and thus I don't have any suggestions.
Author Response
We have read the comments and suggestions from Reviewer #1 and we are confused by the apparent conflicting statements given. In one instance, the reviewer states that we have ignored his/her comments from the first review, and in the other set of comments, he/she seems to imply satisfaction with the changes we have done. With regards to the rest of the form, the reviewer states that the arguments and discussion of findings needs improving. We have restructured and revised the paper following earlier suggestions from the reviewer. The only point we differed with the review was on the value of myths and legends in biodiversity conservation in Africa. The revised submission carries improved arguments and discussions. The references have also been vastly improved.
Reviewer 2 Report
ready to be published
Author Response
Reviewer #2 seems to be satisfied with the revisions we have effected. We thank him/her for the taking the time to review our paper for the second time.